# Ion Channels in Multiple Myeloma: Pathogenic Role and Therapeutic Perspectives

**DOI:** 10.3390/ijms23137302

**Published:** 2022-06-30

**Authors:** Ilaria Saltarella, Concetta Altamura, Aurelia Lamanuzzi, Benedetta Apollonio, Angelo Vacca, Maria Antonia Frassanito, Jean-François Desaphy

**Affiliations:** 1Department of Biomedical Sciences and Human Oncology, Section of Pharmacology, University of Bari Aldo Moro Medical School, Piazza Giulio Cesare 11, I-70124 Bari, Italy; concetta.altamura@uniba.it (C.A.); jeanfrancois.desaphy@uniba.it (J.-F.D.); 2Department of Biomedical Sciences and Human Oncology, Section of Internal Medicine, University of Bari Aldo Moro Medical School, I-70124 Bari, Italy; aurelia.lamanuzzi@uniba.it (A.L.); benedettaapollonio@gmail.com (B.A.); angelo.vacca@uniba.it (A.V.); 3Department of Biomedical Sciences and Human Oncology, Section of General Pathology, University of Bari Aldo Moro Medical School, I-70124 Bari, Italy; antofrassanito@gmail.com

**Keywords:** ion channels, multiple myeloma, drug resistance, therapeutic strategies

## Abstract

Ion channels are pore-forming proteins that allow ions to flow across plasma membranes and intracellular organelles in both excitable and non-excitable cells. They are involved in the regulation of several biological processes (i.e., proliferation, cell volume and shape, differentiation, migration, and apoptosis). Recently, the aberrant expression of ion channels has emerged as an important step of malignant transformation, tumor progression, and drug resistance, leading to the idea of “onco-channelopathy”. Here, we review the contribution of ion channels and transporters in multiple myeloma (MM), a hematological neoplasia characterized by the expansion of tumor plasma cells (MM cells) in the bone marrow (BM). Deregulation of ion channels sustains MM progression by modulating intracellular pathways that promote MM cells’ survival, proliferation, and drug resistance. Finally, we focus on the promising role of ion channels as therapeutic targets for the treatment of MM patients in a combination strategy with currently used anti-MM drugs to improve their cytotoxic activity and reduce adverse effects.

## 1. Introduction

Multiple myeloma (MM) is a hematological malignancy characterized by the aberrant expansion of malignant plasma cells (MM cells) in the bone marrow (BM) [1]. MM develops from the preneoplastic phases, namely monoclonal gammopathy of undetermined significance (MGUS) and smoldering myeloma [2]. The MGUS to MM progression is accompanied by alteration of both MM cells and surrounding BM cells, including fibroblasts, endothelial cells, immune cells, osteoblasts, and osteoclasts that co-evolve to create a tumor-prone niche fostering MM cells’ proliferation, migration, and resistance to apoptosis and cell death. These aberrant mechanisms contribute to tumor onset, angiogenesis, metastasis, immune evasion, and drug resistance [3,4].

Ion channels are pore-forming proteins that allow ions to flow across plasma membranes and cellular organelles in both excitable and non-excitable cells [5]. They are typically involved in the control and modulation of ion homeostasis and secretion, cell excitability, contraction, and the release of neurotransmitters and hormones [6]. Increasing evidence demonstrates the ability of ion channels to affect key biological processes including cell proliferation, differentiation, migration, and apoptosis, all considered “hallmarks of cancers” [7]. Thus, the term “onco-channelopathy” has been recently introduced to indicate the involvement of ion channels and transporters in tumor transformation and progression [8]. Furthermore, ion channels have also been referred to as inducers of drug resistance via the modulation of ion flux and the consequent activation of intracellular pathways [9]. Recently, volume-activated anion channels have been pinpointed as a possible means for chemotherapeutics uptake, such as cisplatin [10,11].

Despite the role of ion channels being deeply investigated in solid tumors [12,13,14,15], less is known about hematological malignancies including MM. Some authors have documented the role of ion channels in activity of B lymphocytes via the regulation of BCR signaling, B cell proliferation, and cytokine release [16], suggesting that a deregulation of ion channels may promote malignant B cell development.

Here, we review the involvement of ion channels and transporters in MM progression and drug resistance by sustaining MM cell survival and proliferation, and by preventing cell death and apoptosis. Finally, we focus on the promising role of ion channels as therapeutic targets for the treatment of MM patients to improve the cytotoxic effect of anti-MM therapies and to prevent their side effects.

## 2. Ion Channels in MM

### 2.1. Potassium Channels

Potassium channels are composed by tetrameric transmembrane proteins. They sustain the resting membrane potential, control the shape and duration of action potential, and participate in excitatory signals, hormone secretion, epithelial function, and cell proliferation [17]. The pleiotropic roles of potassium channels are ensured by the existence of more than 70 K^+^ channel-encoding genes. The K^+^ channels are classified into five different sub-families based on their structure and activation mechanisms [18,19]: (i) voltage-gated (K_v_) channels, the largest subgroup of potassium channels; (ii) two-pore domain (K2P) channels, with two pore domains for each transmembrane subunit; (iii) inward rectifier (Kir) channels, with two transmembrane segments and one extracellular loop that forms the pore [18]; (iv) Ca^2+^- or Na^+^-activated channels (K_Ca_, K_Na_), that are modulated by intracellular variations of calcium or sodium, respectively; and (v) the Slo family, activated by ions (i.e., Na^+^, Cl^−^), pH variations, and phosphorylation [19].

Several studies documented a remodeling of potassium channel expression during tumor onset and progression in solid and hematological malignancies, including MM [20,21,22].

In MM settings, changes in K^+^ currents modify membrane potential and affect intracellular pathways that regulate cell proliferation [23,24,25]. Wang et al. [23] demonstrated that the MM cell line RPMI8226 strongly expresses the voltage-gated K^+^ channels K_v_1.3, whose inhibition with 4-aminopyridine (4-AP) reduces the peak amplitude of the voltage-gated potassium current (Ik_v_), and depolarizes the resting potential. Deregulation of K^+^ current inhibits MM cells’ proliferation by blocking cell cycle progression at the G1 phase and by modulating intracellular pathways’ activation via the regulation of cell volume [23]. Furthermore, the pharmacological treatment with arsenic trioxide (ATO), which decreases Ik_v_ currents and depolarizes cells in a concentration-dependent manner, inhibits MM cell proliferation by arresting RPMI8226 cells at the G0/G1 phase [24]. Wu et al. [25] demonstrated that treatment of MM cells with berberine, an isoquinoline alkaloid, blocks K^+^ currents and reduces MM cell growth. Patch-clamp recordings show that berberine suppresses voltage-activated K^+^ currents, depolarizes resting potential (from +47mV to +39mV), and delays K^+^ channels’ recovery from inactivation. Inside-out configuration indicates that berberine also inhibits Ca^2+^-activated K^+^ channels without changing channel conductance [25].

Genomic alterations affect K^+^ channels’ assembly and function, leading to similar results. A gain of chromosome 1q21, a recurrent mutation observed in MM patients, correlates with overexpression of the Ca^2+^-activated K^+^ channel, K_Ca_2.3, MM progression, and a poor prognosis [26]. A common mutation observed in MM patients involves the 13q14.3 locus that maps the potassium channel regulating gene (KCNRG), whose protein negatively interferes with the K^+^ channel assembly and activity. In vitro studies show that overexpression of KCNRG modifies cell size and shape, leading to a reduction of cell proliferation and migration and a rise of MM cells’ apoptosis, suggesting that deletion of KCNRG may be relevant for MM progression [27].

Overall, these data indicate that the K^+^ channel inhibition induces cell depolarization and modulates cell volume as well as the activity of other ion channels (i.e., Ca^2+^, Na^+^), driving tumor cell proliferation and MM progression (Figure 1) [23].

### 2.2. Chloride Channels

Chloride channels are ubiquitously expressed both in the plasma membrane and in cellular organelles. The cell membrane chloride channels are classified in: (i) voltage-gated chloride channels (CLC); (ii) ligand-gated chloride channels, i.e., γ-aminobutyric acid- (GABA) and glycine-activated receptors; (iii) calcium-activated chloride channels (CaCCs); (iv) cystic fibrosis transmembrane conductance regulator (CFTR); and (v) volume-regulated channels (VRAC) [28]. Recently, chloride intracellular channels (CLICs) have also been identified. They include six proteins (CLIC1-6) mainly expressed on the mitochondrial membrane [29]. Whether they actually function as chloride channels is, however, debated [30].

The chloride channels are involved in the regulation of cell excitability and cell volume, trans-epithelial transport, ion homeostasis, pH regulation, cell migration and infiltration, as well as in the regulation of cell cycle progression. Due to the pivotal role of these functions during malignant transformation, chloride channels are gaining increasing interest for cancerogenesis and tumor progression [31]. Although some authors have demonstrated an altered expression of chloride channels in MM cells (Figure 1) [32,33,34], their role is still little investigated.

Levitan et al. [32] showed the expression of VRAC on RPMI8266 MM cells by analyzing its voltage-dependent inactivation through the whole patch-clamp technique. Next, real-time PCR analysis indicated that MM cells express the chloride channels ClC-3 at higher levels compared to normal plasma cells (PCs). Knock-down of ClC-3 using small interfering RNA (siRNA) reduced IGF-1-induced proliferation and cell cycle progression from the G0/G1 to S phase, with a concomitant reduction of p-ERK1/2, cyclin E, cyclin D1, and an up-regulation of cyclin-dependent kinase (CDK) inhibitors, p21, and p27. Conversely, ClC-3 overexpression sustained MM cell proliferation via the activation of proliferative intracellular pathways, suggesting that ClC-3 may represent an attractive therapeutic target in MM [33].

Recent studies focused on the role of chloride channels in the development of drug resistance in solid and hematological tumors. Zhang et al. [34] showed that the chloride channels ClC-5 may enhance bortezomib resistance of MM cells by activating the pro-survival autophagy. These data corroborate the hypothesis that bortezomib activates pro-survival autophagy in a BM microenvironment [35,36], supporting the importance of ion channels in the development of bortezomib resistance (see “The Role of Ion Channels in Multiple Myeloma Drug Resistance” section).

### 2.3. Calcium Channels and Transporters

Intracellular calcium levels are finely controlled by calcium-selective and non-selective cation channels and transporters: (i) the voltage-activated channels, especially expressed in excitable cells; (ii) the store-operated and second messenger-operated channels (SOCs); (iii) the ligand-gated channels (i.e., the purinergic receptors P2X); and iv) the transient receptor potential (TRP) channel family. Some of these channels, i.e., P2X and TRP, are considered cation channels in that they allow the flux of Na^+^ and/or K^+^ ions (see paragraph “Cation channels”) [37,38]. Moreover, extracellular calcium levels are regulated by a calcium-sensing receptor (CaSR). This is a G protein-coupled receptor that “senses” Ca^2+^ concentration and activates intracellular signaling involved in calcium homeostasis (i.e., re-absorption and/or secretion) [39,40].

As calcium ions are involved in most cellular processes, the aberrant calcium signaling has been related to tumor progression, metastasis, and drug resistance in different tumors, including MM (Figure 1) [41,42].

Approximately 15% of the newly diagnosed MM patients develop hypercalcemia, a pathologic condition that correlates with other myeloma-related comorbidity, i.e., anemia, thrombocytopenia, chromosomal abnormalities, and bone disease [43] and, broadly, with poor prognosis. Since high extracellular levels of Ca^2+^ are present in the BM microenvironment, calcium channels as well as the CaSR may play a pivotal role in the MM pathogenesis. Yamaguchi et al. [44] demonstrated that in vitro exposure of MM cells to high Ca^2+^ levels induces overexpression of the CaSR. Moreover, the stimulation of CaSR by Ca^2+^, polycationic agonists (neomycin sulfate and Gd^3+^), and the specific CaSR activator NPSR467 sustains MM cells proliferation and bone disease [44].

The SOCs include different plasma membrane proteins that enable the Ca^2+^ influx from the extracellular environment as the consequence of calcium depletion in the endoplasmic reticulum (ER). This mechanism, known as store-operated calcium entry (SOCE), is controlled by the stromal-interaction molecule1 (Stim1), a transmembrane protein that monitors for ER Ca^2+^ depletion and activates plasma membrane channels, leading to calcium entry from the extracellular environment [45,46]. The main SOCs channels are represented by Orai1-3, which are unrelated to other channels [47]. Immunohistochemistry analysis of BM biopsies show increased expression of both Stim1 and Orai1 in MM patients compared to healthy subjects. In addition, the Stim1/Orai1 expression increases in step with MM progression and correlates with a shorter survival [48,49]. Inhibition of the Ca^2+^ flux with SOCE inhibitors (SKF-96365, DES, and 2-APB) reduces MM cell viability, blocks cell cycle progression, and induces apoptosis, suggesting that MM cell survival strongly depends on the SOCs’ activity [48]. Yanamandra et al. [50] demonstrated that the farnesyltransferase inhibitor, tipifarnib, induces Ca^2+^ influx from the extracellular space via Orai3, resulting in an increase of cytosolic calcium that leads to ER stress, cell membrane damage and, ultimately, to MM cell death [50]. In addition, SOCE activity is also regulated by the canonical TRP that allows Ca^2+^ influx in response to Ca^2+^ store depletion (see “Cation channels” section) [51].

### 2.4. Non-Selective Cation Channels

The non-selective cation channels allow the flux of both monovalent (Na^+^ and K^+^) and divalent (Ca^2+^ and Mg^2+^) cations. These include the ligand-activated channels (i.e., the purinergic receptors P2X) and the TRP channel family. All these channels are deregulated in MM (Figure 1) [51].

The purinergic receptor P2X7 gated by ATP is one of the most studied in MM. Vangsted et al. [52] analyzed different polymorphisms in the P2X7 gene that produce a gain-of-function (GOF) or loss-of-function (LOF) of P2X7. Specifically, the polymorphism P2RX7:151 + 1 g>t (rs35933842), associated with a complete LOF, correlates with an increased risk to develop MM [52,53]. Nevertheless, Farrell et al. [54] showed that the MM cell line RPMI8226 strongly expresses a functional P2X7 that can be activated by agonists (i.e., ATP and 3′-O-(4-Benzoyl)benzoyl ATP (BzATP)) and inhibited by specific antagonists (i.e., KN-62 and A-438079) as well as by high extracellular levels of Ca^2+^ and Mg^2+^ [54]. The activation of P2X7 induces MM cell death and reduces cell viability by arresting the cell cycle at the S phase through the downregulation of NF-KB signaling [54]. In addition, P2X7 activation reduces bone resorption by modifying the interaction of MM cells with osteoblasts and osteoclasts, suggesting a protective role of P2X7 in myeloma-induced bone disease [55]. These findings are in line with other literature data showing the expression of P2X7 in osteoblasts, osteoclasts, and osteocytes, and its involvement in the regulation of cell proliferation, differentiation, and bone biology [56]. Finally, P2X7 receptors are involved in the induction of neuropathic pain in bortezomib-treated MM patients via the activation of MAPK signaling (see paragraph “Ion channels inhibition in anti-myeloma therapy”).

Another purinergic receptor, P2X5 (also known as LRH-1), is overexpressed in MM cells and in other hematological malignant cells. Overes et al. [57] demonstrated that P2X5-expressing lymphoid tumor cells were recognized and lysed by LRH-1-specific cytotoxic T lymphocytes, suggesting that P2X5 might represent a promising target for immunotherapy.

The TRP channel superfamily encompasses the canonical TRP (TRPC), vallinoid TRP (TRPV), melastatin-related (TRPM), ankyrin TRP (TRPA), mucolipin TRP (TRPML), and polycystic TRP (TRPP) [58]. Growing literature data show that TRP expression is deregulated in hematological malignancies including leukemias, B and T-cell lymphomas, and MM [59,60].

The TRPV2 channel is highly expressed in BM biopsies from MM patients and correlates with bone lesions and to a shorter overall survival, suggesting that its overexpression may represent a marker of poor prognosis. TRPV2 acts as a Ca^2+^-sensing channel. Indeed, exposure of MM cells to high extracellular Ca^2+^ concentrations, which mimics the hypercalcemic condition of BM milieu, promotes TRPV2 expression and induces the release of osteoclastogenic factors including TNFα, IL-1β, and RANKL via activation of the Ca^2+^-calcineurin-NFATc3 pathway. The inhibition of TRPV2 channels by the antagonist SKF96365 in MM cells co-cultured with the monocyte/macrophage-like Raw264.7 cells reduces the expression of the osteoclast markers matrix metalloproteinase 9 (MMP9) and cathepsin K (CTSK), as a sign of decreased MM cells-induced osteoclastogenesis [61], arguing for the relevance of TRPV2 in bone disease.

Bone lesions contribute to create an acidic microenvironment that supports tumor progression, angiogenesis, and genomic instability. Amachi et al. [62] demonstrated that acidic conditions stimulate TRPV1, a pH sensor ion channel that, in turn, activates pro-survival pathways in MM cells including PI3K-AKT, JAK2/STA3, NF-KB, and JNK. Activation of PI3K-AKT enhances the expression of TRPV1 and of the histone deacetylase HDAC1 by inducing the nuclear translocation of the transcription factor Sp1. These data support the existence of paracrine loops that trigger TRPV1 expression and promote MM cells’ survival in acidic conditions. Recently, Beider et al. [63] have further supported the pro-survival role of TRPV1 in MM. Indeed, the inhibition of TRPV1 channels through the antagonist AMG9810 reduces MM cells’ viability by inducing mitochondrial ROS accumulation and affects MM cells’ migration and adhesion by blocking CXCR4/CXCL12 axis. In addition, AMG9810 synergizes with proteasome inhibitors (i.e., bortezomib and carfilzomib) and overcomes bortezomib resistance in MM cells, suggesting that TRPV1 inhibition may represent a new strategy for MM treatment [63] (see “Ion Channels Inhibition as ANTI-myeloma Therapy” section).

Finally, Elzamzamy et al. [49] showed that the anti-MM activity of the cyclic β-hairpin peptide MTI-101, which binds CD44 and induces MM cell death, depends on Ca^2+^/Na^+^ and K^+^ influx via TRPC1/4/5 channels. Indeed, the silencing of STIM1, TRPC1, TRPC4, or TRPC5, as well as the pharmacological inhibition of TRPC activity by SKF96365, reduces MTI-101-mediated Ca^2+^ influx and cell death [49].

### 2.5. Proton Channels

These include highly selective voltage-gated channels (H_v_) that allow H^+^ efflux across the plasma membrane. They have a small conductance and are tightly regulated by membrane potential and pH gradient [64]. By using tight-seal voltage-clamp recording techniques, Schilling et al. proved for the first time the existence of highly-selective H^+^ voltage gated channels in peripheral T and B lymphocytes and in the leukemic cell line Jurkat E6-1 with a high sensitivity to pH variations, membrane depolarization, and to extracellular levels of divalent cations. They demonstrated that the expression of proton channels by immune cells regulates the respiratory burst by sustaining H^+^ extrusion following PMA-induced activation of lymphocytes [65].

As proton channels regulate extracellular pH via H^+^ efflux, they are gaining increasing importance in tumor development for their contribution to the acidic microenvironment and for modulating different biological functions [66]. In breast cancer, H_v_1 channel expression is a negative prognostic factor [67] and promotes tumor cell migration and invasion by activating proteases and MMPs via environment acidification [68]. In chronic lymphocytic leukemia (CLL), malignant B cells overexpress a short form of the H_v_ ion channel that is more sensitive to protein kinase C, resulting in a constitutive activation of BCR signaling and in an increased H^+^ current that sustains tumor cell growth and chemotaxis [69].

The BM microenvironment acidosis is induced by the aberrant expression of proton pumps and carriers [70,71,72]. In MM, the H^+^-ATPase contributes to BM acidification, which mediates bone resorption and pain. MM cells as well as osteoclasts express the plasma membrane a3 isoform of vacuolar H^+^-ATPase (a3V-ATPase) that promotes extracellular acidification and neurite outgrowth of sensory nerves through the activation of the acid-sensing cation channel ASIC3. The activation of ASIC3 induces Ca^2+^ influx and the sprouting of sensory nerves, resulting in bone pain [71]. In addition, BM cells express the Na^+^/HCO3^−^ (NHE10) and the Cl^−^/HCO3^−^ (AE2) exchangers that ensure the H^+^ gradient and extracellular acidification [72].

Overall, these data support the importance of H^+^ channels and transporters as inducers of BM acidification during MM progression and bone metastasis (Figure 1).

### 2.6. Organelle-Related Ion Channels

Different classes of ion channels and transporters are expressed in cell organelles [73]. Voltage-dependent anion channels (VDAC) are the most abundant class of ion channels located on the outer mitochondrial membrane, which regulate mitochondrial permeability to ions and small water-soluble metabolites. Accumulating evidence indicates that VDACs affect mitochondria-mediated apoptosis via protein release through the outer mitochondrial membrane, including cytochrome c [74]. Based on the importance of VDAC in the regulation of apoptosis, several studies investigated their role in tumor progression. Liu et al. [74] demonstrated a positive correlation between VDAC1 and CD45 expression and apoptosis’ susceptibility of MM cells. Indeed, CD45^+^ U266 cells are more sensitive to several apoptotic stimuli, i.e., oxidative stress and ER-stress, compared to CD45^−^ cells. They suggested that intracellular ROS might induce Ca^2+^ release from ER, calcineurin activation, and Bad translocation to mitochondria that leads to cytochrome c release through VDAC1. Accordingly, treatment of CD45^+^ MM cells with thapsigargin, an inhibitor of the sarco/endoplasmic reticulum Ca^2+^ ATPase (SERCA), increases cytosolic levels of Ca^2+^ and triggers apoptosis in VDAC1-expressing MM cells (Figure 1) [75]. Furthermore, Zheng et al. [76] demonstrated that arsenic trioxide (ATO) upregulates VDAC expression and dimerization, resulting in a reduction of mitochondrial membrane potential (ΔΨm) and an increase of MM cell apoptosis through cytochrome c release.

These data emphasize the importance of VDAC in the formation of the mitochondrial permeability transition pores and in the activation of the intrinsic apoptotic pathway. Recently, Kadow et al. [77] showed that MM cells express Kv1.3 channels in their mitochondria, whose targeting might represent a novel anti-MM strategy. Indeed, the inhibition of K^+^ influx by specific Kv1.3 blockers (i.e., PAPTP and PCARBTP) leads to a reduction of respiratory chain activity with the concomitant increase of mitochondrial ROS that culminates into MM cell apoptosis (Figure 1) [77].

Additionally, mitochondria express on their internal membrane the Ca^2+^ uniporter (MCU) channel, a modulator of both cytosolic and mitochondrial calcium levels, which affects cell apoptosis and bortezomib resistance [78,79] (see “Ion channels and drug resistance” section).

The Ca^2+^ homeostasis is also ensured by the Ca^2+^-ATPases that maintain low cytosolic levels of Ca^2+^ and high concentrations of calcium into the ER. The Ca^2+^-ATPases are expressed on the plasma membrane (plasma membrane Ca^2+^-ATPase, PMCA), endoplasmic reticulum (Sarco/endoplasmic reticulum Ca^2+^-ATPase, SERCA), and Golgi apparatus (secretory pathway Ca^2+^-ATPase, SPCA) [80]. Although several studies have documented the involvement of Ca^2+^-ATPases in the regulation of the tumor-associated phenotype [81], the role of Ca^2+^-ATPase in MM still needs to be investigated.

## 3. Ion Channels and Drug Resistance

Although new anti-MM drugs have improved patients’ outcome and overall survival, the MM becomes less responsive to the drugs over time and, ultimately, a relapse occurs [82]. Conventional anti-MM therapies include proteasome inhibitors (bortezomib, carfilzomib, ixazomib), immunomonomodulatory (ImiDs) drugs (lenalidomide, thalidomide, pomalidomide), bruton-kinase inhibitors (ibrutinib), and monoclonal antibodies (daratumumab, elotuzumab) [83]. The ion channels have emerged as key regulators of several biological functions that may mediate drug resistance in hematological tumors including MM [9].

Valverde et al. [84] proposed for the first time the involvement of volume-regulated chloride channels (VRACs) in tumor drug resistance. Others have documented the VRACs role through the reduction of cell apoptosis and/or drug influx [85,86]. Zhang et al. [34] demonstrated that bortezomib treatment significantly upregulates the H^+^/2Cl^−^ exchanger ClC-5 expression in MM cell lines. Overexpression of ClC-5 reduces sensitivity to bortezomib in vitro by activating autophagy, as demonstrated by the increased expression of the autophagic markers, beclin-1 and LC3B-II, and by the modulation of the AKT/mTOR pathway. Accordingly, ClC-5 knockdown decreases the viability of bortezomib-treated MM cells [34].

Bortezomib resistance is also mediated by the mitochondrial MCU channel. The bortezomib-resistant MM cell line KMS20 overexpresses MCU and has an increased membrane potential and high mitochondrial calcium levels compared to the bortezomib-sensitive cell line KMS28BM [78]. Inhibition of the MCU channel with ruthenium red prevents bortezomib-induced apoptosis, supporting the importance of Ca^2+^ regulation via MCU channels as an important mediator of the bortezomib resistance [79].

Moreover, Santoni et al. [87] demonstrated that the mucolipin TRPML2 mediates sensitivity to ibrutinib. Indeed, TRPML2 channel is overexpressed in the ibrutinib-sensitive RPMI8226 cells but not in the ibrutinib-resistant U266 cells. TRPML2 knockdown reduces the anti-MM effect of ibrutinib treatment alone or in combination with bortezomib, suggesting that TRPML2 expression parallels the sensitivity of MM cells to ibrutinib [87].

Furthermore, resistance to MTI-101, the first-in-class peptidomimetic that binds CD44/ITGA4, correlates with a decreased expression of proteins involved in the SOCE pathway and to a reduction of Ca^2+^ flux [88]. Analysis of MTI-101-treated MM cells shows that MTI-101 increases intracellular Ca^2+^ via a release from ER stores and via Ca^2+^ entry through Stim1-mediated activation of TRPC1. These results highlight that MTI-101 induces MM cell death not only by activating the TNFα-Ripk1/Ripk3 necroptotic pathway but also by altering Ca^2+^ flux. Interestingly, MTI-101 has a more potent anti-MM activity in relapsed MM patients, suggesting that relapse is strongly dependent on deregulation of Ca^2+^ homeostasis [88].

The ImiDs exert their anti-MM activity by targeting cereblon, a component of the E3 ubiquitin ligase complex that affects different cell functions by regulating AMPK signaling, glutamine synthetase, as well as the assembly of functional Ca^2+^-activated potassium channels and of voltage-gated chloride channels [89]. Several authors have speculated that ion channels may mediate the anti-MM activity of ImiDs and may represent a marker of response to therapy [89,90].

Overall, these data highlight the crucial role of ion channels in the development of drug resistance to currently used anti-MM drugs, further supporting the possibility to target ion channels in combo strategies to prevent drug resistance.

## 4. Ion Channels Inhibition as Anti-Myeloma Therapy

The ion channels may be envisaged as a new therapeutic target in MM due to their ability to modulate cell survival, proliferation, apoptosis, and drug resistance (Table 1). They can be easily targeted by small molecules as well as by the repurposing of drugs already approved for other channelopathies (i.e., mexiletine, flecainide, acetazolamide) [91].

Different studies demonstrated that the modulation of Ca^2+^ flux synergizes with bortezomib, improving its anti-MM cytotoxic effect [92,93]. Activation of TRPV2 channels by a non-psychoactive cannabinoid has a cytotoxic effect on MM cells. TRPV2 expression by CD138^+^ MM cells correlates with an increased susceptibility to cannabidiol-induced MM cell death. Simultaneous treatment of MM cells with cannabidiol and bortezomib prevents cell growth and cell cycle progression, induces cells death via the activation of ERK, AKT, and NF-κB pathways, and triggers mitochondrial- and ROS-dependent necrosis [92]. In vitro and in vivo studies demonstrated that MTI-101, which exerts anti-MM activity via inhibition of TRPC channels, synergizes with bortezomib and significantly improves mice survival compared to single-agent-treated mice [90]. Moreover, the inhibition of Ca^2+^ influx with tipirfanib enhances the cytotoxic effect of bortezomib, overcoming fibronectin- as well as stroma-mediated drug resistance in vivo [93]. Recently, Beider et al. [63] demonstrated that the inhibition of TRPV1 by using the AMG9810 sensitizes MM cells to bortezomib and overcomes stroma-mediated drug resistance. The simultaneous treatment of AMG9810 and bortezomib interferes with Ca^2+^ homeostasis, induces mitochondrial stress, and affects the ubiquitin pathway in MM cells, ultimately leading to mitophagy and MM cell death. In vivo studies confirmed the synergic anti-myeloma effect of AMG9810 and bortezomib treatment [63].

Finally, as ion channels are important mediators of nerve transmission and nociception, their inhibition may prevent side effects of anti-MM drugs, i.e., peripheral neuropathy, pain, and synaptic and cognitive dysfunction (Table 1) [94]. Neuropathic pain represents one of the major chemotherapy-limiting factors that causes a significant reduction in patients’ quality of life. Promising in vivo studies [95,96] demonstrated that the inhibition of ion channels, including TRPA1 and Cav3.2 T-type Ca^2+^ channels, alleviates bortezomib-induced neuropathic pain. Bortezomib treatment increases both TRPA1 and the IL-6 receptor in the dorsal root ganglion of rats. Inhibition of the TRPA1 channel as well as of the IL-6 pathway reduces mechanical and cold sensitivity of treated rats. The disruption of IL-6 signaling decreases intracellular activation of p38-MAPK and JNK pathways, as well as TRPA1 expression in the sensory neuron, suggesting a role of IL-6 in mediating neuropathic pain via the induction of TRPA1 expression [95]. Similarly, Li et al. [96] described that bortezomib-induced neuropathy is also sustained by TNF-α through the activation of intracellular pathways that induce TRPA1 expression. These data highlight the involvement of IL-6/TRPA1 and TNF-α/TRPA1 axis in bortezomib-induced neuropathy.


ijms-23-07302-t001_Table 1Table 1Therapeutic perspectives of modulation of ion channel activity with agonist and/or antagonist/inhibitor molecules for the treatment of MM patients.ChannelsIon Channels ModulatorMolecular EffectsBiological EffectsCombination StrategyRef.TRPV2CannabidiolActivation of ERK, AKT and NF-κBInhibits cell growth and cell cycle progression,induces cell death and mitochondrial- and ROS-dependent necrosisBortezomib[92]SOCsMTI-101-Improves cell and mice survivalBortezomib[88]Ca^2+^ channelsTipirfanib-Overcomes fibronectin- and stroma-mediated drug resistanceBortezomib[93]TRPA1HC030031Activation of p38-MAPK and JNKAlleviates bortezomib-induced neuropathic painBortezomib[95,96]Cav3.2 T-typecalcium channels(2R/S)-6-prenylnaringenin, KTt-45 and TTA-A2-Alleviates bortezomib-induced neuropathic painBortezomib[97]TRPM8Menthol(agonist)-Alleviates bortezomib-induced neuropathic painBortezomib[98]TRPV1Capsaicin(agonist)-Mediates thalidomide-induced analgesia and cognitive dysfunctionThalidomide[99]
AMG9810Inhibition of ubiquitin pathwayMitochondrial ROS accumulation, mitophagy and MM cells deathBortezomib[63]Calcium-activatedK+ (BK) channelsPaxillineReduces K+ conductanceAlleviates thalidomide-induced cognitive dysfunctionThalidomide[100]


Bortezomib treatment triggers overexpression of the Cav3.2 T-type calcium channel by preventing their proteasome-mediated degradation, resulting in an increase of Ca^2+^ influx that mediates nerve transmission and neuropathic pain [97]. Conversely, based on previous findings that demonstrated the analgesic effect of cooling compounds that activate TRPM8 [101,102], Colvin et al. [98] described a case report of a MM patient with a grade IV bortezomib-induced neuropathic pain treated with menthol to activate TRPM8 channel and to successfully reverse peripheral neuropathy.

Other studies demonstrated that ion channels mediate thalidomide-induced analgesia and cognitive dysfunction, i.e., impairment of spatial association, reduced recognition memory, inhibition of working memory, and depression [99,100]. In vitro and in vivo studies show that thalidomide reduces capsaicin-mediated activation of TRPV1 channels and cation influx [98]. As TRPV1 mediates pain sensation, thalidomide administration induces an analgesic effect by attenuating nociceptive pain. Finally, Choi et al. [99] investigated the cognitive dysfunctions observed in thalidomide-treated MM patients using the C57BL6 mouse model in vivo. Analysis of mice behavior showed that the thalidomide-induced cognitive deficits parallel an increased expression and activity of calcium-activated K^+^ (BK) channels in mice hippocampus. Accordingly, inhibition of calcium-activated K^+^ channels with paxilline reduces K^+^ conductance and alleviates cognitive symptoms [100].

Overall, these studies shed light on the mechanisms involved in the side effects of anti-MM therapy and propose ion channels-based combination strategies for the treatment of MM patients.

## 5. Concluding Remarks and Future Perspectives

Increasing evidence documents the important role of ion channels in the regulation of several biological processes involved in tumor onset, progression, and drug resistance, leading to the concept of “tumor as a channelopathy”.

Accordingly, ion channels are now considered a promising target for the treatment of oncological diseases. Surface ion channels activity may be easily modulated by small molecules and/or drugs already approved for the treatment of other diseases, suggesting the possibility of drug repurposing for the treatment of solid and hematological cancers. However, as tumor deregulated ion channels are also expressed by normal cells, the identification of tumor-specific isoforms would be pivotal to increase sensitivity and specificity of ion channels-based therapy in cancer, preventing the adverse effects of ion channel targeting [103].

## Figures and Tables

**Figure 1 ijms-23-07302-f001:**
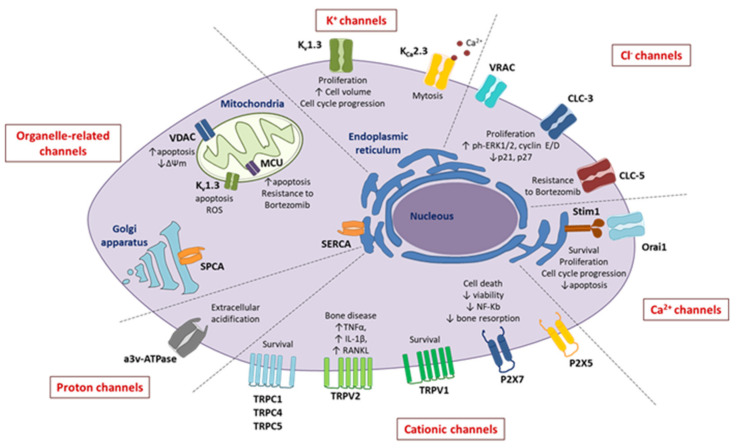
Schematic representation of ion channels/exchangers expressed in MM cells, which may be involved in MM pathogenesis and drug resistance.

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
