# Peer review of "Ion Channels in Multiple Myeloma: Pathogenic Role and Therapeutic Perspectives"

_ijms, 2022, doi:10.3390/ijms23137302_

Round 1
Reviewer 1 Report
I think that the review of Saltarella I, et al "Ion channels in multiple myeloma: pathogenic role and therapeutic perspectives" can be accepted for publication. The authors have covered this topic extensively. The Figure 1 shows all the ion channels that are involved in multiple myeloma. I recommend that you revise the list of references to include articles published in recent years and request that the follow article be included in the review : Beider K et al. J Hematol Oncol 2020, 13: 53. PMID: 33239060
Author Response
The authors thank the Reviewer 1 for helpful criticism and are glad for positive comments.
Reviewer ’s comment:
I recommend that you revise the list of references to include articles published in recent years and request that the follow article be included in the review: Beider K et al. J Hematol Oncol 2020, 13: 53. PMID: 33239060.
Reply: We thank the Reviewer for this comment. Accordingly, we have included and discussed the manuscript by Beider K et al. as follow: “Recently, Beider et al. [63] have further supported the pro-survival role of TRPV1 in MM. Indeed, inhibition of TRPV1 channels through the antagonist AMG9810 reduces MM cells viability by inducing mitochondrial ROS accumulation and affects MM cells migration and adhesion by blocking CXCR4/CXCL12 axis. In addition, AMG9810 synergizes with proteasome inhibitors (i.e. bortezomib and carfilzomib) and overcomes bortezomib re-sistance in MM cells suggesting that TRPV1 inhibition may represent a new strategy for MM treatment [62] (see “Ion channels inhibition as anti-myeloma therapy” section).” (page 6, lines 237-244) and “Recently, Beider et al. [63] demonstrated that inhibition of TRPV1 by using the AMG9810 sensitizes MM cells to bortezomib and overcomes stroma-mediated drug resistance. The simultaneous treatment of AMG9810 and bortezomib interferes with Ca2+ homeostasis, induces mitochondrial stress and affects the ubiquitin pathway in MM cells, ultimately leading to mitophagy and MM cell death. In vivo studies confirmed the synergic anti-myeloma effect of AMG9810 and bortezomib treatment [63].” (page 8, lines 383-389). Table 1 was accordingly updated.
In addition, according to Reviewer’s suggestion to revise references list, we have also inserted a very recent article published on April 28th by Omari et al. as reference 60.
As the new references 60 by Omari et al. and 63 by Beider et al. have been inserted, all the following references have been renumbered.

Reviewer 2 Report
In this manuscript, the authors summarized the roles of ion channels in multiple myeloma. These ion channels include potassium channels, chloride channels, calcium channels, non-selective cation channels, proton channels, and transporters. Specifically, they talked about how the ion channels are involved in the development of multiple myeloma, in terms of expression and function. They also showed that some ion channels play a role in the drug resistance of anti-multiple myeloma. Lastly, the inhibition of ion channels is promising for anti-myeloma treatment. The review is well-written and clear.
Author Response
The authors are glad for Reviewer 2 positive comments.
